# Cardiomyopathy and Sudden Cardiac Death: Bridging Clinical Practice with Cutting-Edge Research

**DOI:** 10.3390/biomedicines12071602

**Published:** 2024-07-18

**Authors:** Raffaella Mistrulli, Armando Ferrera, Luigi Salerno, Federico Vannini, Leonardo Guida, Sara Corradetti, Lucio Addeo, Stefano Valcher, Giuseppe Di Gioia, Francesco Raffaele Spera, Giuliano Tocci, Emanuele Barbato

**Affiliations:** 1Department of Clinical and Molecular Medicine, Sapienza University of Rome, 00189 Rome, Italy; armando.ferrera@uniroma1.it (A.F.); luigi.salerno3@gmail.com (L.S.); fvannini2907@gmail.com (F.V.); leonardo.guida@uniroma1.it (L.G.); sara.corradetti@gmail.com (S.C.); francesco0802@gmail.com (F.R.S.); giuliano.tocci@uniroma1.it (G.T.); emanuele.barbato@uniroma2.it (E.B.); 2OLV Hospital Aalst, 9300 Aalst, Belgium; addeolucio@gmail.com (L.A.); stefano.valcher@gmail.com (S.V.); 3Department of Advanced Biomedical Sciences, University of Naples Federico II, Corso Umberto I, 40, 80138 Naples, Italy; 4Cardiovascular Department, Humanitas University, Via Alessandro Manzoni, 56, 20089 Rozzano, Italy; 5Institute of Sports Medicine and Science, National Italian Olympic Committee, Largo Piero Gabrielli, 1, 00197 Rome, Italy; dottgiuseppedigioia@gmail.com

**Keywords:** cardiomyopathies, sudden cardiac death, cardiovascular prevention, risk stratification

## Abstract

Sudden cardiac death (SCD) prevention in cardiomyopathies such as hypertrophic (HCM), dilated (DCM), non-dilated left ventricular (NDLCM), and arrhythmogenic right ventricular cardiomyopathy (ARVC) remains a crucial but complex clinical challenge, especially among younger populations. Accurate risk stratification is hampered by the variability in phenotypic expression and genetic heterogeneity inherent in these conditions. This article explores the multifaceted strategies for preventing SCD across a spectrum of cardiomyopathies and emphasizes the integration of clinical evaluations, genetic insights, and advanced imaging techniques such as cardiac magnetic resonance (CMR) in assessing SCD risks. Advanced imaging, particularly CMR, not only enhances our understanding of myocardial architecture but also serves as a cornerstone for identifying at-risk patients. The integration of new research findings with current practices is essential for advancing patient care and improving survival rates among those at the highest risk of SCD. This review calls for ongoing research to refine risk stratification models and enhance the predictive accuracy of both clinical and imaging techniques in the management of cardiomyopathies.

## 1. Introduction

Sudden cardiac death (SCD) prevention in cardiomyopathies, such as hypertrophic (HCM) (OMIM #192600), dilated (DCM) (OMIM #115200), non-dilated left ventricular (NDLCM), and arrhythmogenic right ventricular cardiomyopathy (ARVC) (OMIM #107970), remains a crucial but complex clinical challenge, especially among younger populations [1]. Accurate risk stratification is hampered by the variability in phenotypic expression and genetic heterogeneity inherent in these conditions. This article explores the multifaceted strategies for preventing SCD across a spectrum of cardiomyopathies and emphasizes the integration of clinical evaluations, genetic insights, and advanced imaging techniques such as cardiac magnetic resonance (CMR) in assessing SCD risks. Advanced imaging, particularly CMR, not only enhances our understanding of myocardial architecture but also serves as a cornerstone for identifying at-risk patients. The integration of new research findings with current practices is essential for advancing patient care and improving survival rates among those at the highest risk of SCD [2,3].

## 2. Materials and Methods

A comprehensive literature search was conducted using the PubMed database to identify relevant studies on cardiomyopathies (HCM, DCM, NDLCM, and ARVC) and sudden cardiac death (SCD). The search included articles published from January 2000 to December 2023. The keywords used were “hypertrophic cardiomyopathy”, “dilated cardiomyopathy”, “non-dilated left ventricular cardiomyopathy”, “arrhythmogenic right ventricular cardiomyopathy”, “sudden cardiac death”, “risk stratification”, “implantable cardioverter-defibrillator”, and “cardiac magnetic resonance”. Additional articles were identified through manual searches of reference lists from key articles and reviews. Only studies published in English were considered. Studies were included if they met specific criteria: they had to be original research articles, reviews, or meta-analyses focused on any of the specified cardiomyopathies (HCM, DCM, NDLCM, and ARVC) and their association with sudden cardiac death. They needed to discuss risk factors, genetic insights, and advanced imaging techniques, particularly cardiac magnetic resonance (CMR), and provide data on clinical outcomes and risk stratification models. We excluded studies that were not related to the specified cardiomyopathies or sudden cardiac death, case reports and small case series with fewer than 10 patients, and non-English publications. A narrative synthesis of the findings was performed, focusing on the integration of clinical, genetic, and imaging data in the prevention of sudden cardiac death in patients with cardiomyopathies.

## 3. Hypertrophic Cardiomyopathy: Incidence and Risk Factors for Sudden Cardiac Death

HCM is a genetic (autosomal dominant) heart muscle disease caused by a mutation in sarcomere protein genes characterized by an increased left ventricular (LV) wall thickness (>14 mm) or mass that is not solely explained by abnormal loading conditions. It can be considered familial when two or more first- or second-degree relatives with HCM or a first-degree relative with autopsy-proven HCM and sudden death at <50 years of age are detected [1,4]. Once considered rare, it is now recognized as the most prevalent genetic heart condition. Determined from echocardiographic studies, the prevalence of HCM in the adult population is 0.2%, whilst in children it is 0.029% [5,6].

HCM is an important cause of heart failure and atrial fibrillation (AF) [4,7], even if the most common cause of death in these patients is arrhythmic SCD mediated primarily by ventricular fibrillation [8,9]. SCD in HCM often affects young and frequently asymptomatic patients [6]. The annual SCD rate is <1%, but within the general population with HCM but there are subgroups with a much higher incidence.

Since ventricular fibrillation (VF) appears to be the principal mechanism of sudden death in patients with HCM, in high-risk patients implantable defibrillators (ICDs) are highly effective in terminating such arrhythmias, indicating that these devices are the gold standard for primary and secondary prevention of sudden death [10]. Drugs are not effective in primary or secondary prevention of SCD in HCM patients [11].

Therefore, assessing the risk of SCD is crucial in clinical management. The 2022 European Society of Cardiology (ESC) guidelines advise that for secondary prevention of SCD in HCM patients, ICD implantation is recommended for all individuals experiencing haemodynamically unstable ventricular tachycardias (VTs) or ventricular fibrillation (VF) (class I, level of evidence B). Additionally, in patients with HCM who have haemodynamically stable sustained monomorphic VT, ICD implantation is suggested (class IIa, level of evidence C) [12].

Risk stratification and primary prevention of SCD in HCM is more challenging. Over several decades, a multitude of studies have focused on the identification of major clinical risk markers that stratify patients according to level of risk to identify high-risk patients who may be candidates for SCD primary prevention with ICDs. This risk stratification strategy and the penetration of ICDs into clinical practice has substantially reduced disease-related mortality rates.

For this reason, SCD risk assessment at the initial visit and repeated every 1 to 2 years is a critical part of the evaluation of patients with HCM [13].

A large body of evidence suggests that on the basis of history taking, imaging evaluation, and continuous (24/48 h) ambulatory electrocardiographic monitoring, the most important established risk markers are as follows: age, unexplained syncope, extreme left ventricular wall thickness, HCM-related sudden death in a first-degree relative, and multiple or prolonged episodes of non-sustained ventricular tachycardia (NSTV), left atrial size, and LV outflow tract (LVOT) gradient [14,15].

LV systolic dysfunction, LV apical aneurysm, as well as extensive late gadolinium enhancement (LGE) on CMR have been added to the risk-stratification algorithm for HCM [16].

Some of these risk factors are widely recognized by both major HCM guidelines (American and European), while others are considered differently.

### 3.1. Age

SCD is more common in younger patients, especially those under the age of 35 years [17].

Until recently, primary prevention data for children were scarce, before the advent of specific scores and risk calculators. The HCM Risk-Kids score, designed and externally validated for children aged 1–16 with HCM, includes factors such as unexplained syncope, maximum LV wall thickness, large left atrial diameter, low LVOT gradient, and non-sustained ventricular tachycardia (NSVT). Unlike in adults, adding age and family history of SCD did not improve the predictive accuracy of the pediatric model [18,19].

### 3.2. Unexplained Syncope

The 2017 American Heart Association (AHA)/American College of Cardiology (ACC) and the 2018 European Society of Cardiology (ESC) syncope guidelines defined unexplained syncope as “syncope for which a cause is undetermined after an initial evaluation that is deemed appropriate by the experienced healthcare provider [20,21]”.

In a recent systematic review and metanalysis, syncope was reported by 15.8% of HCM patients and life-threatening arrhythmic events occurred in 3.6% of non-syncopal patients and in 7.7% of syncopal patients during a mean follow-up of 5.6 years (relative risk of 1.99). Syncope was considered unexplained in 91% of cases [22,23].

There are numerous factors that might lead to syncope in patients suffering from HCM. These factors include hypovolemia, complete atrioventricular block, sinus node dysfunction, sustained ventricular tachycardia, obstruction of the left ventricular outflow tract, atrial arrhythmias with a rapid ventricular response, and abnormal vascular reflexes [14].

Unexplained syncope has long been recognized as a marker for heightened risk of sudden death; nevertheless, the timing of syncopal episodes in relation to patient evaluation has emerged as clinically significant. Recent instances of unexplained syncope were linked to a heightened risk of sudden death across all age brackets when compared to patients without syncope. This finding suggests that recent unexplained syncope could warrant consideration for the prophylactic implantation of ICD. Conversely, remote episodes of syncope did not correlate with increased risk in older patients [24].

Additionally, patients with recurring episodes of unexplained syncope, who are at low risk of SCD, should be considered for an implantable loop recorder [25].

### 3.3. Maximum LV Wall Thickness

In HCM, the magnitude of hypertrophy is directly related to the risk of sudden death and is a strong and independent predictor of prognosis. In most clinically diagnosed cases, left ventricular wall thickness is 15 mm or more (average, 21 mm), but there is massive thickness (30 to 50 mm) in some cases [26].

Elliot et al. [27] showed the prognostic significance of LV hypertrophy in relation to other clinical risk factors: 630 patients with HCM were evaluated, and it was observed that there was a trend towards a greater likelihood of sudden death or ICD discharge with increasing wall thickness (especially in those with a wall thickness of 30 mm or more). However, it was shown that the estimated risk of sudden death or ICD discharge was influenced more by the number of clinical risk factors than by the extent of wall thickness. In patients with a wall thickness of 30 mm or more, the estimated risk of sudden death or ICD discharge at 5 years ranged from 5% in patients with no other risk factors to 34% in patients with three additional risk factors.

In another study, Olivotto et al. observed that only in patients diagnosed at a very young age might the presence of extreme LV wall thickness represent, per se, a potential marker of risk of sudden death and that the degree of maximum LV wall thickness should be considered in the context of a multifactorial approach to risk stratification, rather than as an isolated risk factor [28].

### 3.4. NSVT

NSVT is defined as three consecutive ventricular beats at a rate of ≥120 beats per minutes and <30 s in duration during Holter monitoring (minimum duration 24 h) at or prior to evaluation [29].

Monserrat et al. studied 531 HCM patients undergoing Holter monitoring and found that around 20% of these patients had at least one episode of NSVT [28].

In patients aged ≤30 years (but not in those who are >30 years old), freedom from sudden death at five years was lower in patients with NSVT and there was also no relationship between the duration, frequency, or rate of NSVT episodes and prognosis at any age [30].

Moreover, Gimeno et al. showed that in a cohort of 1380 patients with HCM the five-year survival from sudden death or ICD discharge was significantly lower in patients with exercise-induced NSVT or VF [HR 3.14 (95% CI: 1.29–7.61, *p* = 0.01)] [31].

### 3.5. Family History of SCD

History of SCD is defined as SCD in one or more first-degree relatives under 40 years of age, or SCD in a first-degree relative with confirmed HCM at any age (diagnosed post-mortem or ante-mortem) [32].

Compared with HCM patients without an obvious family history, patients with a family history of SCD had a 20% increased risk of SCD [33]. The recognition that SCD can affect multiple members within the same family, combined with clinical research showing that a family history of sudden death related to HCM independently predicts sudden death, underscores the importance of considering family history as a significant risk factor. In suitable clinical contexts, this family history can serve as the basis for recommending prophylactic ICD therapy [33]. According to ACC/AHA guidelines, in HCM patients with a family history of SCD, ICD implantation should be recommended (Class IIa, level of evidence B) [34].

### 3.6. LVOT Gradient

The presence of a peak LVOT gradient of ≥30 mmHg is considered to be indicative of obstruction, while resting or provoked gradients of ≥50 mmHg are generally considered the threshold for septal reduction therapy in patients with drug-refractory symptoms [34].

The maximum LVOT gradient should be determined at rest and with Valsalva provocation (irrespective of concurrent medical treatment) using pulsed- and continuous-wave Doppler from the apical three and five chamber views. The peak outflow tract gradient should be determined using the modified Bernoulli equation, gradient = 4V², where V is the peak aortic outflow velocity [15].

Elliot et al. [27] showed that LVOT obstruction is associated with an increased risk of SD/ICD [91.4% (95% CI: 87.4–95.3) vs. 95.7% (95% CI: 93.8–97.6), *p* = 0.0004] that is related to the severity of obstruction [RR per 20 mmHg = 1.36 (95% CI: 1.12–1.65), *p* = 0.001] and the presence of other recognized risk factors for SD.

Moron et al. investigated the impact of LVOT obstruction on morbidity and mortality in a substantial cohort of HCM patients. The study involved the evaluation of 1101 patients, revealing a notably elevated likelihood of death associated with HCM among those with outflow tract obstruction compared to those without (RR 2.0; *p* = 0.001). Consequently, left ventricular outflow tract obstruction at rest was deemed a robust independent predictor of mortality [35].

### 3.7. Left Atrial Size

The left atrial diameter, quantified with echocardiography, has been associated with SCD in HCM patients. The left atrial diameter should be determined by M-mode or 2D echocardiography in the parasternal long axis plane at the time of evaluation [15]. The left atrial size may reflect the risk of SCD, as it may relate to atrial remodeling secondary to increasing ventricular fibrosis, which makes the myocardium more susceptible to arrhythmias [17].

### 3.8. Additional Risk Factors: LV Systolic Dysfunction, LV Apical Aneurysm, and Extensive LGE on CMR

HCM with LV systolic dysfunction (LVSD) is defined as occurring when left ventricular ejection fraction is <50%. LVSD affects around 8% of patients with HCM.

Although the natural history of HCM with LV systolic dysfunction was variable, 75% of patients experienced adverse events, including 35% experiencing a death equivalent in an estimated median time of 8.4 years after developing systolic dysfunction [36].

AHA/ACC guidelines consider LVSD a major risk factor for SCD in HCM patients, recommending ICD implantation in patients with HCM and a reduced ejection fraction (<50%) (class IIa, level of evidence C), whilst the ESC guidelines decided to use the presence of a reduced LV ejection fraction (<50%) only as an additional value in the decision-making process [1,21].

LV apical aneurysms are outpouchings of the left ventricular apex, described as discrete, thin-walled dyskinetic or akinetic segments of the most distal portion of the left ventricular chamber, that are relatively common in patients with apical HCM or HCM with midventricular obstruction [37].

The frequency of this occurrence in patients selected without bias is unclear, but they were documented in 3% of individuals in the prospective Hypertrophic Cardiomyopathy Registry [38].

This represents a non-negligible category of patients within the clinical spectrum of HCM. This finding raises a number of management considerations, including risk stratification for SCD.

The areas of myocardial scarring contiguous to the scarring edge of the aneurysm, at the junction between vital and abnormal tissue, represent the site where re-entry circuits occur, thus constituting the primary arrhythmogenic substrate for the generation of malignant ventricular tachyarrhythmias, irrespective of the size of the aneurysm [39]. However, VT may be repetitive, thereby raising considerations for additional treatment strategies such as radiofrequency ablation [40].

Rowin et al. retrospectively analyzed 1940 consecutive HCM patients, 93 of whom (4.8%) were identified to have an apical LV aneurysm. In the study, the authors point out that HCM patients with apical aneurysm had an adverse event rate of 6.4%/year, more than three times higher than the HCM cohort without aneurysm. Around 20% of the patients with aneurysms in this group underwent potentially life-saving ICD interventions for ventricular tachycardia/ventricular fibrillation. In nearly half of the patients, an ICD was implanted mainly due to the presence of the aneurysm itself. This results in an arrhythmic event rate of nearly 5% per year, which is more than five times higher than that observed in our patient cohort without aneurysms [41].

Another study by Papanastasiou et al. [42] also showed that left ventricular apical aneurysm in HCM patients is associated with an increased risk of SCD events (pooled OR: 4.67, 95% CI: 2.30 to 9.48, I2: 38%).

Based on these and other studies, to date there is a difference in the treatment pathways of SCD risk assessment. AHA/ACC guidelines include LV aneurysms as a major independent SCD risk factor and they are considered a reasonable sole indication for an ICD (class IIa, level of evidence C) [34].

Conversely, the ESC guidelines [1] suggest that the number of events is insufficient to determine the independent predictive value of apical aneurysms in SCD. For instance, in the study by Rowin et al. [39], most SCD events were ICD interventions for monomorphic VT, indicating a significant inclusion bias. Additionally, many subjects with events had other critical risk markers, such as previous sustained ventricular arrhythmias. Based on the current data, the European Task Force recommends that decisions on ICD implantation for primary prevention of SCD should rely on well-established risk factors rather than solely on the presence of an apical left ventricular aneurysm [1]. Recently, LGE during CMR has become an in vivo marker for myocardial fibrosis, although its role in stratifying the sudden death risk in HCM subgroups is not yet fully understood. Chan et al., investigated the relationship between LGE and cardiovascular outcomes in 1293 HCM patients undergoing CMR, with an average follow-up of 3.3 years. They discovered that an LGE extension of 15% or more of LV mass was associated with a twofold increase in the risk of SCD events in patients otherwise considered at lower risk, with an estimated 5-year SCD event probability of 6%. Conversely, the absence of LGE correlated with a lower risk of SCD events [43].

Several meta-analyses [44,45,46] have confirmed these data, showing that LGE is common in HCM and that, when extensive (expressed as a percentage of LV mass), it is associated with an increased risk of SCD and other events, particularly in the presence of other markers of disease severity.

In addition to the arbitrary cutoff of ≥15% of LV mass (exclusive of right ventricular insertion areas) a linear relationship is demonstrated between sudden death risk and LGE extension, suggesting that an LGE of 10–15% can be clinically relevant in some patients; absent or focal LGE (<5% of LV mass) is generally regarded as most consistent with low risk [47].

AHA/ACC guidelines suggest that extensive LGE on CMR in HCM patients may be an indication for ICD implantation (class IIb, level of evidence C), whilst ESC guidelines do not recommend ICD implantation in primary prevention for SCD only on the basis of the presence of an extensive LGE at CMR in HCM patients [48] Figure 1.

## 4. Dilated Cardiomyopathy: Incidence and Risk Factors for Sudden Cardiac Death

DCM is characterized by the enlargement of the left ventricle and a decrease in its ability to pump effectively, without any blockages in the coronary arteries or other adverse conditions affecting the heart’s workload. While some studies suggest that DCM may affect as many as one in 250 individuals, precise contemporary prevalence figures are not readily available [49]. Despite advancements in treatment over the last few decades that have led to better survival rates, DCM continues to pose a significant global health threat, primarily due to heart failure and SCD. The incidence of SCD in a large registry of unselected DCM patients has fallen from 2% per year before current therapy to 0.15% per year now [50,51].

ICDs can identify and promptly address life-threatening heart rhythms, reducing the risk of SCD. However, determining which patients will benefit most from ICD therapy involves a complex evaluation balancing the individual’s arrhythmic risk against the risk of death from other causes. The approach of using a single measurement of the LVEF and NYHA class to gauge the relative risk of SCD is widely acknowledged as an inadequate method for determining ICD candidacy [49]. Registry data reveal a subset of patients with an LVEF > 35% in whom a notable proportion of SCD cases occur, indicating the need to consider additional factors when evaluating the appropriateness of ICD implantation in a disease characterized by significant etiological diversity [52,53,54]. Multi-parametric assessment of myocardial, electrical, and genetic substrates can predict significant arrhythmias. Balancing the risk of non-sudden death allows for personalized therapy and prevents wasteful device placement for people who may not benefit.

### 4.1. Clinical Characteristics

ICD implantation for primary prevention of SCD is currently recommended for patients with DCM who have a left ventricular ejection fraction (LVEF) ≤ 35%, New York Health Association (NYHA) class II or III symptoms, and who have been treated with optimal therapy for at least 3 months, with a life expectancy of >1 year [1].

However, research looking into the use of ICDs for primary SCD prevention in DCM patients has yielded mixed results. Key trials like The Sudden Cardiac Death in Heart Failure Trial (SCD-HeFT) and Defibrillators in Non-Ischemic Cardiomyopathy Treatment Evaluation (DEFINITE) have contributed significantly to understanding ICD efficacy [52,53]. For instance, SCD-HeFT enrolled patients with both ischemic cardiomyopathy and DCM, showing an all-cause mortality benefit in those receiving single-chamber ICDs [52,55]. In contrast, the DEFINITE trial, focusing on DCM patients, did not find a significant reduction in overall mortality in the ICD group compared to OMT (HR 0.65; 95% CI 0.40–1.06; *p* = 0.08) in spite of a significant reduction in SCD (HR 0.20; 95% CI 0.06–0.71; *p* = 0.006) [53]. The DANISH study sheds light on the critical features of predicting SCD and risk stratification for ICD therapy in current clinical practice. For starters, it highlights the rarity of life-threatening arrhythmic events in this population, as indicated by the low frequency of SCD in the control group. This finding is consistent with a larger examination of individuals with systolic heart failure, which showed a decrease in SCD with time due to improved guideline-based management [56]. Additionally, the potential effects of new treatments, such as angiotensin receptor–neprilysin inhibitors (ARNIs) and sodium–glucose cotransporter 2 (SGLT2) inhibitors, on reducing this risk have yet to be formally evaluated. It is important to note that both age and comorbidities seem to influence the risk of SCD in DCM. A predetermined analysis of the DANISH trial indicated that age significantly impacts the effectiveness of ICDs on overall mortality. Notably, patients aged 70 or younger have a higher incidence of SCD, with a greater proportion of their deaths attributed to SCD compared to older patients. In these younger patients, ICDs have been shown to reduce overall mortality by 30%. Conversely, in patients over 70, the occurrence of non-sudden deaths is double that of younger patients, which diminishes the impact of ICDs on reducing overall mortality [55]. Furthermore, the presence of other health conditions can increase the risk of non-sudden death, affecting the overall benefit derived from ICDs [57,58]. A meta-analysis of data from three primary prevention ICD trials revealed that a decline in kidney function, measured by glomerular filtration rate, correlated with a reduced survival benefit from ICDs in patients with heart failure with reduced ejection fraction (HFrEF), including those with DCM [59]. Additionally, analysis from the SCD-HeFT trial found that patients with type 2 diabetes, despite facing a higher risk of SCD, did not see a reduction in SCD or overall mortality with ICD use, irrespective of whether their heart failure was ischemic or non-ischemic [60]. These insights suggest that while comorbidities play a crucial role in influencing the arrhythmic risk in DCM, their impact on the effectiveness of ICD therapy needs further investigation. Furthermore, advanced risk assessment models like the Seattle Heart Failure Model and the Seattle Proportional Risk Model, which include a range of comorbidities, demonstrate promise in distinguishing between risks of SCD and death from progressive heart failure in HFrEF patients, including those with DCM. However, the practical application of these models in predicting the benefits of ICD implantation remains to be fully validated in clinical settings [61,62].

### 4.2. Genetic Background

Recent studies indicate that phenotype influences the risk of SCD. Patients with variants in PLN, DSP, LMNA, FLNC, TMEM43, DES, and RBM20 have a significantly higher incidence of major arrhythmic events compared to other causes of DCM, regardless of LVEF values [63,64]. Laminopathy is linked to an especially severe condition marked by high penetrance, early onset of ventricular arrhythmias, an atrioventricular (AV) block, and progression to advanced heart failure. The combination of a high risk of ventricular arrhythmias and the potential for bradyarrhythmia due to an AV block places patients with LMNA mutations at a particularly elevated risk of sudden cardiac death (SCD). This necessitates a much lower threshold for considering the implantation of an implantable cardioverter–defibrillator (ICD) [65]. In a recent retrospective study involving 1161 patients diagnosed with DCM, it was found that individuals carrying pathogenic or likely pathogenic (P/LP) causative genetic variants faced a more challenging clinical trajectory and exhibited a higher incidence of major arrhythmic events compared to those without identified genetic variants. This trend was particularly pronounced when comparing them to patients with DCM and an LVEF ≤ 35%. Additionally, regardless of their left ventricular ejection fraction (LVEF) levels, patients with DCM and specific causative genetic variants were observed to have an elevated risk of major arrhythmic events. Notably, genes associated with a heightened risk of arrhythmias included those encoding nuclear envelope proteins (LMNA, EMD, and TMEM43), desmosomal proteins (DSP, DSG2, DSC2, and PKP2), and certain cytoskeleton proteins. These findings collectively suggest that DCM patients harboring pathogenic variants in high-risk genes (such as LMNA, EMD, TMEM43, DSP, RBM20, and PLN, and truncating variants of FLNC) should be identified as having a predisposition to SCD [66]. Therefore, the consideration of ICD placement for primary prevention should extend beyond conventional LVEF threshold values >35%, especially in the presence of additional risk factors (e.g., non-sustained ventricular tachycardia, frequent premature ventricular contractions, male gender, significant late gadolinium enhancement, and specific genetic variants). For selecting high-risk genotypes (e.g., LMNA), gene-specific (or variant-specific, as seen with the p.Arg14del variant of PLN) risk estimation scores have been developed to incorporate genotype and additional phenotypic characteristics [67,68]. Whenever available, these risk scores should inform the decision-making process regarding ICD implantation for primary prevention.

### 4.3. Late Gadolinium Enhancement

Patients with DCM who do not possess a high-risk genotype or who have an LVEF greater than 35% might benefit from risk stratification based on the presence and extent of myocardial scarring detected by LGE on CMR. Myocardial replacement fibrosis, identifiable through late gadolinium enhancement cardiovascular magnetic resonance imaging (LGE-CMR), is found in around 30% of DCM patients, typically situated in the mid-wall of the septum. Numerous studies have established a strong association between the presence of non-ischemic LGE and SCD [69]. In a significant prospective observational cohort study by Gulati et al. [70], which tracked 472 DCM patients over a median of 64 months, a strong connection was observed between the presence of LGE on CMR and the composite arrhythmic endpoint (SCD or aborted SCD), even after adjusting for the left ventricular ejection fraction (LVEF) [71]. Furthermore, a less potent association was noted with all-cause mortality and a composite heart failure endpoint. Subsequent evaluation of the dose–response relationship between LGE and SCD risk in a larger DCM patient cohort revealed a non-linear correlation, suggesting that the mere presence of LGE might be a better predictor of risk than its extent alone. This study also highlighted the heightened risk of SCD or aborted SCD in patients with both septal and left ventricular free wall LGE [71]. A prospective study focusing on patients with a milder DCM phenotype and no existing indication for an ICD demonstrated LGE’s predictive power for SCD or aborted SCD in individuals with an LVEF > 40%, underscoring its significance even in patients not meeting current guidelines for primary prevention ICD placement [72]. Recent progress in computational modeling techniques has shed light on the varied fibrosis patterns and densities observed with late gadolinium enhancement (LGE) in patients with dilated cardiomyopathy (DCM), associating these patterns with the potential for re-entry and arrhythmogenesis. Furthermore, an alternative method that evaluates LGE entropy, which measures the heterogeneity of scarring, has shown independent predictive value for major arrhythmic events in a registry-based study of DCM patients with primary prevention ICDs [73]. In conclusion, the additional insights provided by LGE assessment offer significant prognostic value beyond that of echocardiography, making it the preferred imaging technique for assessing DCM patients.

### 4.4. Markers of Electrical Instability

Additional considerations, including syncope, the presence of NSVT, and the frequency of ventricular extrasystoles (VEs), are also potential indicators to assist in determining the need for ICD implantation. Currently, there is insufficient data to establish a precise threshold for VE burden, as it may vary depending on the patient’s genotype and other clinical variables [74]. For patients experiencing unexplained syncope, programmed electrical stimulation (PES) may offer further insights into the underlying etiology. Current European guidelines include syncope in the algorithm for ICD, because these events are mostly related to ventricular tachycardias [75].

### 4.5. Future Perspectives

Cardiovascular magnetic resonance parametric mapping provides a non-invasive method for assessing diffuse interstitial fibrosis in various cardiac conditions. Puntmann et al. [76] found a correlation between native T1 values and both all-cause mortality and composite heart failure outcomes. Additionally, there may be a link between diffuse fibrosis and arrhythmogenesis, as T1 mapping was predictive of major arrhythmias in patients with ischemic cardiomyopathy and DCM. However, further research is needed to validate this technique and determine its additional value alongside LGE imaging.

Looking ahead, diffusion tensor cardiovascular magnetic resonance (DT-CMR) emerges as a promising technique for the non-invasive examination of cardiac microstructure at the cellular level [77]. While studies have revealed microstructural abnormalities in DCM patients, the potential of DT-CMR in predicting SCD remains unexplored. Notably, a study on HCM patients demonstrated an association between low fractional anisotropy and ventricular arrhythmia, suggesting a possible new imaging biomarker for arrhythmic risk. Further investigation into this relationship in DCM patients could provide valuable insights into microstructural pathophysiology and arrhythmogenesis [78].

The convergence of advanced cardiac imaging with artificial intelligence and machine learning holds significant promise. These approaches have already been utilized for CMR image analysis, offering opportunities for extracting vast datasets and developing new risk markers in DCM patients. Ongoing studies are exploring the potential utility of these applications [79].

Also, left ventricular global longitudinal strain (LV GLS) serves as a reliable alternative to the left ventricular ejection fraction (LVEF) for assessing LV contractile function and may be more sensitive to subtle dysfunction. Myocardial strain has shown associations with survival across various cardiovascular conditions. Romano et al. documented a link between LV GLS and all-cause mortality in a large cohort of patients with an LVEF < 50%, indicating that each 1% decline in GLS corresponded to an 89.1% increase in the risk of death after adjusting for clinical and imaging variables. However, the correlation between LV GLS and ventricular arrhythmias remains uncertain, as indicated by a retrospective echocardiography-based study. It is unclear whether myocardial strain can effectively identify candidates for ICD therapy or if it is more adept at stratifying heart failure mortality rather than arrhythmic events [80] (Figure 2).

## 5. Arrhythmogenic Right Ventricular Cardiomyopathy: Incidence and Risk Factors for Sudden Cardiac Death

ARVC is a genetically determined heart disease that predisposes individuals to SCD, particularly in young patients and athletes, characterized anatomically by fibrofatty replacement of the myocardium and clinically by prominent ventricular arrhythmias and impairment of ventricular systolic function. ARVC has a prevalence that varies between 1/1000 and 1/5000 subjects in the general population, and the variability can be explained by the difficulty on the one hand of detecting the disease, especially when a mild form is present, and on the other hand by SCD as the first manifestation [81]. A meta-analysis of 52 cohort studies (total of 5485 ARVC patients) reported an incidence of SCD of ∼0.7% per year in ARVC patients without an ICD, and a significantly lower SCD incidence (0.65 per 1000 patients per year) in ICD recipients [82]. However, some post-mortem studies have documented the presence of ARVC in 20–30% of SCD victims and in 10–13% of young athletes with SCD [83].

### 5.1. Clinical Characteristics

Clinical data indicate that extensive myocardial involvement in ARVC leads to a greater risk of severe ventricular arrhythmias and SCD. Factors such as right ventricular dilatation and systolic dysfunction, particularly a reduced fractional area change, left ventricular involvement, overt heart failure, and signs of advancing structural disease are linked to an increased SCD risk [84]. Recent episodes of syncope (within 6–12 months) have similarly been tied to a heightened SCD risk [85]. Studies also suggest that men with ARVC are more likely to experience ventricular arrhythmias and SCD than women, potentially due to higher levels of physical activity and specific biological mechanisms [86]. Moreover, high-intensity physical activity has been associated with earlier onset of symptoms, increased risk of ventricular arrhythmias, and a greater likelihood of progressive heart failure leading to the need for heart transplantation. Consequently, high-intensity exercise is generally discouraged for individuals with ARVC, including those who carry the gene but do not show symptoms [87].

### 5.2. Genetic Background

ARVC is typically an inherited condition linked to pathogenic gene variants mainly in desmosome-related genes, including *PKP2* (plakophilin-2), *DSP* (desmoplakin), *DSG2* (desmoglein-2), and *DSC2* (desmocollin-2), though occasionally in non-desmosome genes as well [88]. The genetic variants generally display an autosomal dominant inheritance pattern with incomplete penetrance, except for the fully penetrant TMEM43 p.S358L variant, which is linked to progressive cardiomyopathy and a substantial arrhythmic risk. Individuals with mutations tend to experience an earlier onset of the disease [89]. Notably, possessing multiple mutations correlates with a higher risk of early ventricular arrhythmias and a worse overall prognosis [90]. Approximately 30% of family members show disease penetrance, and positive genetic testing can pinpoint individuals at an elevated risk of developing ventricular arrhythmias. Nonetheless, there is some disagreement regarding the independent predictive value of genetic factors for SCD in ARVC. It is important to highlight that specific non-desmosomal mutations, such as those in *TMEM43*, *LMNA*, and *PLN*, are particularly associated with a high risk of SCD [91].

### 5.3. Markers of Electrical Instability

Patients with ARVC frequently display electrocardiographic abnormalities, such as T-wave inversions in the right precordial leads (usually V1 to V3), reduced QRS voltages in the limb leads, and a terminal activation delay referred to as the ‘epsilon wave’ in the right precordial leads [92]. A history of sustained ventricular tachycardia (VT), typically with a left bundle branch block pattern, is recognized as a significant predictor of increased SCD risk in individuals with ARVC. Furthermore, studies have linked additional markers of electrical instability, such as NSVT and frequent ventricular premature complexes, to an increased SCD risk [93,94,95].

### 5.4. Future Perspectives: Myocardial Fibrosis and Fatty Infiltration

Recent findings suggest that CMR imaging provides crucial prognostic details in cases of arrhythmogenic right ventricular cardiomyopathy (ARVC). Notably, signs of myocardial fibrosis and fatty infiltration identified through CMR are linked to a heightened risk of serious adverse events such as SCD or aborted SCD, and the necessity for appropriate ICD interventions [96]. Since 2005, the prognostic significance of late gadolinium enhancement (LGE) in arrhythmogenic cardiomyopathy (ACM) patients has been well recognized. Tandri et al., through a study comparing the results of contrast-enhanced CMR with electrophysiological studies and endomyocardial biopsies, found that RV LGE presence was associated with the inducibility of arrhythmias during electrophysiological tests [97]. A more recent piece of research by Aquaro et al. involved a comprehensive CMR assessment incorporating wall motion, chamber size, function, and tissue characterization of both ventricles. The study classified patients according to their phenotypic presentation into categories of isolated right ventricular (RV) disease, isolated left ventricular (LV) disease, biventricular disease, and absence of structural disease [98]. The findings indicated that distinct ACM CMR phenotypes are associated with different prognostic outcomes, especially highlighting that patients with left ventricular late gadolinium enhancement (LV LGE) had a worse prognosis compared to those with only right ventricular disease. Furthermore, recent research has identified that elevated native T1 and extracellular volume (ECV) values serve as predictors of negative outcomes during follow-up [99] (Figure 3).

## 6. Non-Dilated Left Ventricular Cardiomyopathy

NDLVC is characterized by non-ischemic left ventricular (LV) scarring or fatty deposits identified through CMR, without enlargement of the LV. This condition may also present with either global or regional LV wall motion irregularities, or as an isolated global LV hypokinesia (i.e., LVEF < 50%) that cannot be explained by abnormal loading conditions. NDLVC has a genetic basis similar to that of DCM and arrhythmogenic right ventricular cardiomyopathy (ARVC), as depicted in Figure 1. Most of the knowledge regarding the natural progression and SCD risk associated with NDLVC comes from research on patients with DCM and ARVC. The presence of high-risk genetic markers such as *LMNA, TMEM43, DSP, RBM20, PLN,* and *FLNC* truncating variants is a primary factor determining the risk of SCD in NDLVC. Currently, there is a lack of dependable risk assessment methods for patients without identified gene mutations. Consequently, until more data are available, the existing guidelines recommend that the approach to primary ICD implantation for patients with NDLVC should align with those established for patients with DCM [1,10] (Figure 4).

## 7. Conclusions

The prevention of SCD in cardiomyopathies requires a multifaceted approach combining genetic screening, clinical assessment, and advanced imaging. Our review highlights the critical role of CMR in identifying structural abnormalities that precede SCD, particularly in conditions like HCM and arrhythmogenic right ventricular cardiomyopathy (ARVC).

Our findings, along with the literature, suggest that factors such as extreme left ventricular hypertrophy, myocardial fibrosis detected by LGE, and genetic predispositions significantly impact SCD risk. By integrating these factors, we can better stratify risk and guide the use of preventive measures like ICDs.

This approach aims to improve the prediction and prevention of SCD, enhancing patient outcomes through tailored management strategies.

## Figures and Tables

**Figure 1 biomedicines-12-01602-f001:**
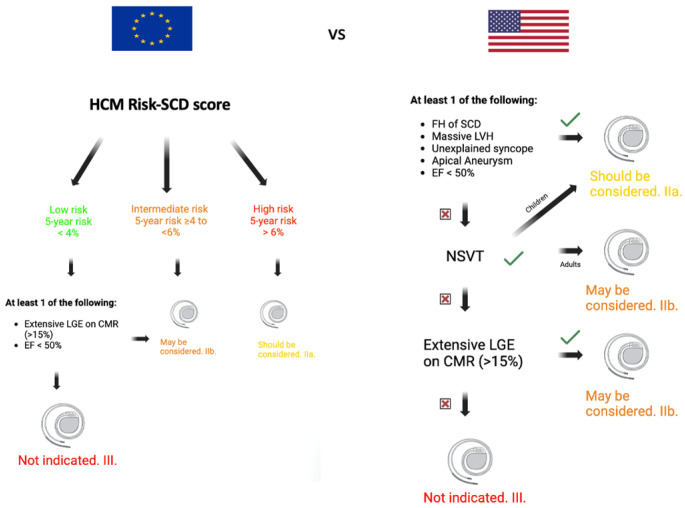
Comparison between European and American guidelines for SCD prevention in HCM patients. CMR: cardiac magnetic resonance; EF: ejection fraction; FH: family history of sudden cardiac death; LVH: left ventricular hypertrophy; LGE: late gadolinium enhancement; and NSVT: non-sustained ventricular tachycardia.

**Figure 2 biomedicines-12-01602-f002:**
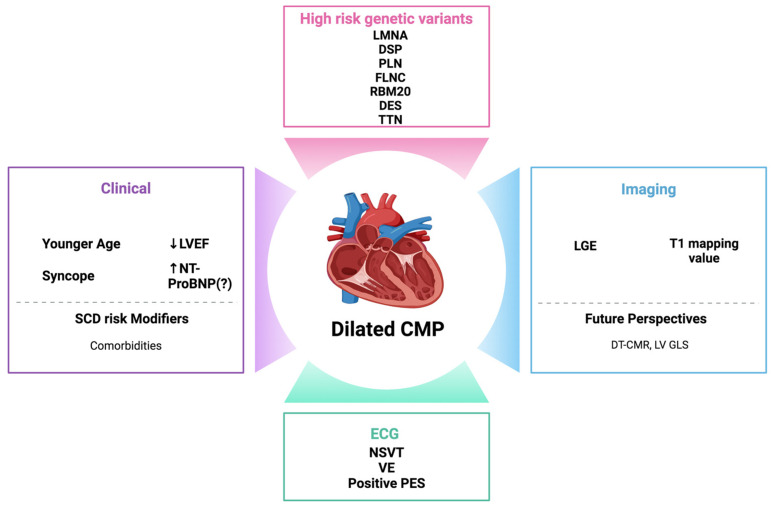
Established and emerging risk factors for sudden cardiac death in dilated cardiomyopathy. LMNA, lamin A/C; DSP, desmoplakin; PLN, phospholamban; FLNC, filamin C; RBM20, RNA-binding motif protein 20; DES, desmin; TTN, titin; LGE, late gadolinium enhancement; ECG, electrocardiogram; NSVT: non-sustained ventricular tachycardia; VE, premature ventricular complexes; PES, programmed electrical stimulation; LVEF, left ventricular ejection fraction; NT-proBNP, N-terminal pro-B-type natriuretic peptide; and SCD, sudden cardiac death.

**Figure 3 biomedicines-12-01602-f003:**
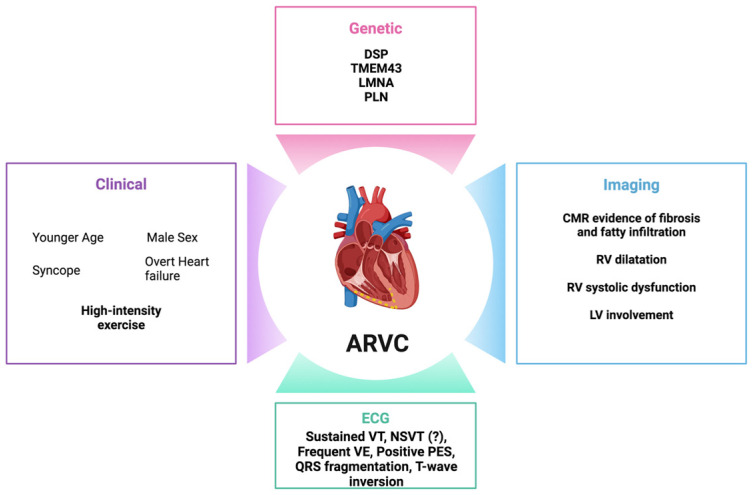
Established and emerging risk factors for sudden cardiac death in arrhythmogenic right ventricular cardiomyopathy. DSP, desmoplakin; TMEM43, transmembrane protein 43; LMNA, lamin A/C; PLN, phospholamban; CMR, cardiac magnetic resonance; LV, left ventricular; RV, right ventricular; ECG, electrocardiogram; NSVT; non-sustained ventricular tachycardia; VE, premature ventricular complexes; VT, ventricular tachycardia; and PES, programmed electrical stimulation.

**Figure 4 biomedicines-12-01602-f004:**
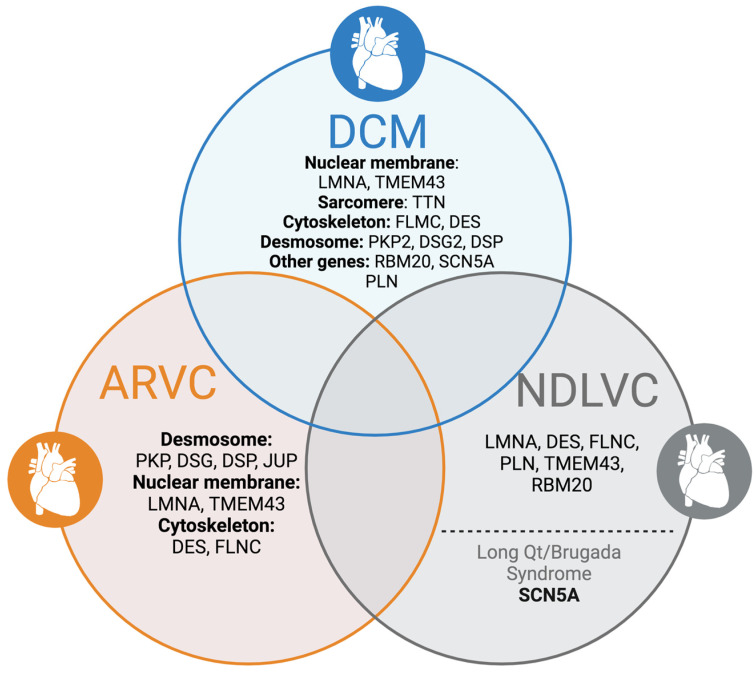
Overlapping genetic background in cardiomyopathies. * The list is not exhaustive. ARVC, arrhythmogenic right ventricular cardiomyopathy; DCM, dilated cardiomyopathy; DES, desmin; DSG, desmoglein; DSP, desmoplakin; FLNC, filamin C; JUP, plakoglobin; LMNA, lamin A/C; NDLVC, non-dilated left ventricular cardiomyopathy; PKP, plakophilin; PLN, phospholamban; TMEM43, transmembrane protein 43; RBM20, RNA-binding motif protein 20; and SCN5A, sodium channel protein type 5.

## Data Availability

No new data were created or analyzed in this study.

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
