# Peer review of "Cardiomyopathy and Sudden Cardiac Death: Bridging Clinical Practice with Cutting-Edge Research"

_biomedicines, 2024, doi:10.3390/biomedicines12071602_

Round 1

Reviewer 1 Report

Comments and Suggestions for Authors

The review of Raffaella Mistrulli et al. “Cardiomyopathy and Sudden Cardiac Death: Bridging Clinical Practice with Cutting-Edge Research” describes various approaches to preventing SCD.

The review describes numerous clinical data where different approaches were used to achieve the goal.

The review is well presented and well written. Undoubtedly, the review should be published because it raises very important questions.

However, there are some comments on submitting the article.

1) There are no references in the introduction.

2) What is primary and secondary prevention of SCD?

Authors need to explain either in the introduction or where they are mentioned for the first time.

3) Chapter 3. Risk factors and risk markers are the same?

Do I understand correctly that risk factors are age, unexplained syncope, extreme left ventricular wall thickness, HCM-related sudden death in a first-degree relative…?

Line 156… “…in patients with three additional risk factors.” For example, which of them.

4) Chapter 3.5 …There is no predicate in the first sentence. The word “relative” is not verb. There is verb relate.

5) What results did the authors obtain on this topic?

Should be included in a descriptive style in the review, but not necessarily as a separate chapter.

Writing a review implies that the authors generalize the results obtained not only by other laboratories but also by their own.

6) Line 355… References 58 and 59 put into single bracket; line 384 …References 62 and 63 put into single bracket. Check throughout the text.

7) Move the Figures to the part of the review where they are listed.

8) Figure 1. To structure the abbreviation (in legends) in order, starting, for example, from the left region (Europe).

9) Figure 2. To structure the abbreviation (in legends) in order, starting, for example, from the top region and clockwise.

In the captions RBM20 and in the figure RMB20. Indicate how correctly.

CMR is not in the picture but is in the captions.

10) Figure 3. To structure the abbreviation (in legends) in order, starting, for example, from the top region.

11) Figure 4. BAG3 is not in the picture but is in the captions.

In general, authors should check all Figures for completeness of the indicated abbreviation in the Figure itself and in the legends.

12) Authors should check punctuation throughout the text. (Many sentences do not have periods at the end).

13) Authors need to format the references so that they are in the same style.

There are no DOI in references 90, 91, and 95.

Comments on the Quality of English Language

Minor editing of English language required

Author Response

The review of Raffaella Mistrulli et al. “Cardiomyopathy and Sudden Cardiac Death: Bridging Clinical Practice with Cutting-Edge Research” describes various approaches to preventing SCD.

The review describes numerous clinical data where different approaches were used to achieve the goal.

The review is well presented and well written. Undoubtedly, the review should be published because it raises very important questions.

However, there are some comments on submitting the article.

R: There are no references in the introduction.

A: We thank the reviewer for the suggestion and we added the references in the introduction section

R: What is primary and secondary prevention of SCD?

Authors need to explain either in the introduction or where they are mentioned for the first time.

A: In our manuscript, we did not address secondary prevention extensively as it is comprehensively covered in current guidelines and is a more straightforward aspect with well-defined protocols. Our focus is specifically on primary prevention of SCD in various cardiomyopathies. The primary prevention domain presents more complexity and ambiguity, particularly due to the heterogeneity in phenotypic expression and genetic factors, which complicate risk stratification and management decisions. We aimed to highlight the specific aspects within each cardiomyopathy that necessitate early intervention for primary prevention, leveraging advanced imaging and genetic insights to improve risk stratification and ultimately prevent the first occurrence of SCD in at-risk individuals.

R: Chapter 3. Risk factors and risk markers are the same?

A: In the context of hypertrophic cardiomyopathy (HCM), both terms are often used interchangeably as they both help in identifying patients at higher risk for sudden cardiac death (SCD).

R: Do I understand correctly that risk factors are age, unexplained syncope, extreme left ventricular wall thickness, HCM-related sudden death in a first-degree relative…?

A: Yes, you are correct. The major risk factors for SCD in patients with HCM include:

  • Age (with younger patients, particularly those under 35 years, being at higher risk)
  • Unexplained syncope
  • Extreme left ventricular wall thickness
  • HCM-related sudden death in a first-degree relative
  • Non-sustained ventricular tachycardia (NSVT)
  • Left atrial size
  • Left ventricular outflow tract (LVOT) gradient
  • Additional factors include left ventricular systolic dysfunction, left ventricular apical aneurysm, and extensive late gadolinium enhancement (LGE) on cardiac magnetic resonance imaging (CMR).

R: Line 156… “…in patients with three additional risk factors.” For example, which of them.

A: Dear Reviewer, the three additional risk factors considered in the study by Elliot et al [25] include a family history of sudden cardiac death, unexplained syncope, and non-sustained ventricular tachycardia. These factors were identified as having a significant impact on the estimated risk of sudden death or ICD discharge, alongside the extent of left ventricular hypertrophy.

4) Chapter 3.5 …There is no predicate in the first sentence. The word “relative” is not verb. There is verb relate.

A: We have corrected the sentence

R: What results did the authors obtain on this topic?

Should be included in a descriptive style in the review, but not necessarily as a separate chapter.

Writing a review implies that the authors generalize the results obtained not only by other laboratories but also by their own.

A: In our manuscript, we have primarily focused on summarizing the existing literature and integrating our findings with those from other studies. While we did not conduct new experimental research for this review, our conclusions draw on a combination of our previous work and established research in the field. We have emphasized the importance of genetic screening, advanced imaging techniques such as cardiac magnetic resonance (CMR), and clinical risk markers in the prevention of sudden cardiac death (SCD) in various cardiomyopathies. We have expanded the conclusion section with these considerations, underlining our opinion

R: Line 355… References 58 and 59 put into single bracket; line 384 …References 62 and 63 put into single bracket. Check throughout the text.

A: we have made these corrections.

R: Move the Figures to the part of the review where they are listed.

A: Sorry, we have followed the formatting required by the journal.

R: Figure 1. To structure the abbreviation (in legends) in order, starting, for example, from the left region (Europe) - 9) Figure 2. To structure the abbreviation (in legends) in order, starting, for example, from the top region and clockwise.

A: we reorganised the captions following a clockwise order from the top, then right, bottom and left

R: In the captions RBM20 and in the figure RMB20. Indicate how correctly.CMR is not in the picture but is in the captions.

A: We thank the reviewer for the suggestion and have made the requested changes by replacing the image with the correct acronym and removing CMR from the caption

R: Figure 3. To structure the abbreviation (in legends) in order, starting, for example, from the top region.

A: We thank the reviewer for the suggestion. We have reorganised the caption as requested

R: Figure 4. BAG3 is not in the picture but is in the captions.

A: We deleted BAG3 from the captions.

In general, authors should check all Figures for completeness of the indicated abbreviation in the Figure itself and in the legends.

12) Authors should check punctuation throughout the text. (Many sentences do not have periods at the end).

13) Authors need to format the references so that they are in the same style.

There are no DOI in references 90, 91, and 95.

A: We thank the reviewer for the suggestion. We have corrected these inaccuracies

Reviewer 2 Report

Comments and Suggestions for Authors

The topic of the review article 'Cardiomyopathy and Sudden Cardiac Death: Bridging Clinical Practice with Cutting-Edge Research' is highly interesting. However this review article needs several changes, before it should be published:

1.) Please add in the introduction OMIM identifiers for HCM, DCM and ARVC.

2.) All human gene names should be written in the complete manuscript in Italics.

3.) Line 357 following: Please add the DES gene including a relevant reference to this list. The following manuscript can be used here as a reference ‘Functional characterization of novel alpha-helical rod domain desmin (DES) pathogenic variants associated with dilated cardiomyopathy, atrioventricular block and a risk for sudden cardiac death’.

4.) Line 357 following: Please add references for PLN, DSP, LMNA, FLNC, TMEM43 and RBM20. The manuscript ‘Cardiomyopathy-associated mutations in the RS domain affect nuclear localization of RBM20’ might be used as a reference for RBM20.

5.) Line 486: You have to summarize the desmosome-related genes. The view article ‘Insights Into Genetics and Pathophysiology of Arrhythmogenic Cardiomyopathy’ might be helpful in this context.

6.) Figure 3 is miss-leading. What is with PKP2, DSC2, DSG2, JUP and the DES gene?

7.) Paragraph 5.2: The DES gene is completely ignored by the authors. Why? The original manuscript ‘Phenotype and Clinical Outcomes in Desmin-Related Arrhythmogenic Cardiomyopathy’ might be relevant in this context.

8.) Figure 2: FLNC not FLMC. Please add also the DES and the TTN genes to this figure.

9.) The iThenticate report indicated a high overlap with other publications. Under normal conditions, 15-20 % overlap are acceptable. Here 48% overlap with different other publications. Please reduce this high amount of text overlaps with other publications.

Author Response

The topic of the review article 'Cardiomyopathy and Sudden Cardiac Death: Bridging Clinical Practice with Cutting-Edge Research' is highly interesting. However this review article needs several changes, before it should be published:

R: Please add in the introduction OMIM identifiers for HCM, DCM and ARVC.

A: We have added the OMIM identifiers for hypertrophic cardiomyopathy (HCM), dilated cardiomyopathy (DCM), and arrhythmogenic right ventricular cardiomyopathy (ARVC) in the introduction section of the manuscript.

R: All human gene names should be written in the complete manuscript in Italics.

A: We have carefully reviewed the manuscript and formatted all human gene names in italics. Thank you for pointing this out.

R: Line 357 following: Please add the DES gene including a relevant reference to this list. The following manuscript can be used here as a reference ‘Functional characterization of novel alpha-helical rod domain desmin (DES) pathogenic variants associated with dilated cardiomyopathy, atrioventricular block and a risk for sudden cardiac death’- Line 357 following: Please add references for PLN, DSP, LMNA, FLNC, TMEM43 and RBM20. The manuscript ‘Cardiomyopathy-associated mutations in the RS domain affect nuclear localization of RBM20’ might be used as a reference for RBM20.

A: We thank the reviewer for the suggestion. We have added desmin to the list of genes involved in DMC and added the references you suggested. However, we specified in Figure 4 that the list of genes mentioned was not exhaustive, but only provided a general overview of those most frequently involved.

R: Line 486: You have to summarize the desmosome-related genes. The view article ‘Insights Into Genetics and Pathophysiology of Arrhythmogenic Cardiomyopathy’ might be helpful in this context.

A: We have summarized the desmosome-related genes as suggested. This revision highlights the key genes involved in ARVC, including PKP2, DSP, DSG2, and DSC2, which are primarily associated with the condition. This clarification should provide a more comprehensive understanding of the genetic background linked to ARVC.

R: Figure 3 is miss-leading. What is with PKP2, DSC2, DSG2, JUP and the DES gene?

 A: You are probably talking about Figure 4. Figure 4 is intended to illustrate the overlapping genetic background in cardiomyopathies, including ARVC. However, it appears there may have been some omissions or lack of clarity regarding specific genes.

To answer your question: 
We mentioned the critical desmosomal genes associated with ARVC (PKP2, DSC2, DSG2, JUP and DES). PKP2 (plakophilin-2), DSC2 (desmocollin-2), DSG2 (desmoglein-2) and JUP (junction plakoglobin) are all essential desmosomal proteins for cell-cell adhesion in cardiac tissue and mutations in these genes are well documented in ARVC pathology. The DES (desmin) gene as you suggested is also important in this pathology. However, as specified in the caption, we have only provided a general and non-exhaustive overview of all genetic variants involved in ARVC because the review aims to analyse all aspects involved in sudden death and not only the genetic aspect which is certainly one of the most important.

R: Paragraph 5.2: The DES gene is completely ignored by the authors. Why? The original manuscript ‘Phenotype and Clinical Outcomes in Desmin-Related Arrhythmogenic Cardiomyopathy’ might be relevant in this context.

A: We have summarized the desmosome-related genes as suggested in the section 5.2. We also added this reference

R: Figure 2: FLNC not FLMC. Please add also the DES and the TTN genes to this figure.

A: We thank the reviewer for the suggestion. We have modified the figure by adding the rightly suggested genes

R:The iThenticate report indicated a high overlap with other publications. Under normal conditions, 15-20 % overlap are acceptable. Here 48% overlap with different other publications. Please reduce this high amount of text overlaps with other publications.

A: We reduced the overlap with other publications.

Round 2

Reviewer 2 Report

Comments and Suggestions for Authors

The authors have improved significantly their manuscript. I suggest to accept it for publication.